

# The *pht4;1-3* mutant line contains a loss of function allele in the *Fatty Acid Desaturase 7* gene caused by a remnant inactivated selection marker—a cautionary tale

Anders K. Nilsson and Mats X. Andersson

Department of Biology and Environmental Sciences, University of Gothenburg, Gothenburg, Sweden

## ABSTRACT

A striking and unexpected biochemical phenotype was found in an insertion mutant line in the model plant *Arabidopsis thaliana*. One of two investigated insertion mutant lines in the gene encoding the phosphate transporter PHT4;1 demonstrated a prominent loss of trienoic fatty acids, whereas the other insertion line was indistinguishable from wild type in this aspect. We demonstrate that the loss of trienoic fatty acids was due to a remnant inactive negative selection marker gene in this particular transposon tagged line, *pht4;1-3*. This constitutes a cautionary tale that warns of the importance to confirm the loss of this type of selection markers and the importance of verifying the relationship between a phenotype and genotype by more than one independent mutant line or alternatively genetic complementation.

## INTRODUCTION

Reverse genetics is a powerful tool in plant biology to establishing causal relationships between genotype and phenotype. There are numerous mutagenesis strategies, both targeted (*Felippes, Wang & Weigel, 2012*; *Gaj, Gersbach & Barbas, 2013*; *Yin, Gao & Qiu, 2017*) and untargeted (*Alonso et al., 2003*; *Henikoff & Comai, 2003*; *McCallum et al., 2000*) for generating the starting material for reverse genetics studies. Once the genetic location of a mutation has been confirmed, and the corresponding gene product is shown to be affected, the work in connecting genotype to phenotype can begin (*O'Malley & Ecker, 2010*). But importantly, the plant line must also be cleared of off-target mutations that may have arisen during mutagenesis. The common recommendation to circumvent this problem is to use several independent mutant lines when inferring phenotype from genotype, or alternatively, the mutant line can be functionally complemented with the wild-type copy of the gene of interest.

The *PHT4;1* gene in *Arabidopsis thaliana* (At2g29650) encodes a phosphate transporter localized to chloroplast membranes (*Guo et al., 2008*; *Karlsson et al., 2015*; *Pavon et al., 2008*; *Yin, Vener & Spetea, 2015*). Several loss-of-function mutant alleles for PHT4;1

Corresponding authors
Anders K. Nilsson,
anders.nilsson@bioenv.gu.se
Mats X. Andersson,
mats.andersson@bioenv.gu.se

have been described: *pht4;1-1*, *pht4;1-2* and *pht4;1-3* (*Wang et al., 2011*). The *pht4;1-3* allele was isolated from a transposon insertion mutant population (*Sundaresan et al., 1995*; *Wang et al., 2011*) and has been used in at least two publications (*Guo et al., 2008*; *Karlsson et al., 2015*). The PHT4;1 transporter has not only been reported to be involved in maintaining chloroplast phosphate homeostasis (*Karlsson et al., 2015*) but also to play a role in plant pathogen defense by affecting salicylic acid levels (*Wang et al., 2014*; *Wang et al., 2011*). Further, plants devoid of PHT4;1 display reduced growth under standard cultivation condition, a phenotype that can be reverted by growing the *pht4;1* mutants in high-phosphate conditions (*Karlsson et al., 2015*).

The membranes of plant chloroplast are mainly comprised of galactolipids, of which mono- and digalactosyl diacylglycerol (MGDG and DGDG, respectively) make up approximately 75% of the total chloroplastic acyl lipids (*Li-Beisson et al., 2013*). MGDG and DGDG are particularly enriched in trienoic fatty acids with 16 or 18 carbons in chain length (16:3 and 18:3, respectively, number of total carbon atoms:number of double bonds of the fatty acids). Arabidopsis mutants defective in the *Fatty acid desaturase* (*FAD7*) gene show reduced plastidial conversion of linoleic acid (18:2) to $\alpha$-linolenic acid (18:3), and hexadecadienoic acid (16:2) to hexadecatrienoic acid (16:3) (*Browse, McCourt & Somerville, 1986*; *Iba et al., 1993*). Hence, *fad7* plants have a fatty acid profile skewed to contain unusually high proportion of 16:2/18:2–16:3/18:3.

Under phosphate limiting conditions, plants can exchange a substantial part of the plasma membrane phosphoglycerolipids for galactolipids (*Andersson et al., 2003*). In light of these previous findings, we investigated whether *pht4;1* mutants are affected in leaf lipid composition. To our surprise we identified a second mutation in the *pht4;1-3* line located in the *FAD7* locus that rendered plants defective in the synthesis of trienoic fatty acids. We propose that this phenotype, not connected to *PHT4;1 per se*, could potentially influence properties previously ascribed to absence of the phosphate transporter.

## MATERIALS AND METHODS

### Plant material, growth conditions and genotyping

Arabidopsis plants were cultivated as described under short day condition (*Johansson et al., 2015*). The *pht4;1-2* and *pht4;1-3* lines were a kind gift from professor Cornelia Spetea Wiklund and have been previously described (*Karlsson et al., 2015*). The *fad7-1* knock-out line (NASC ID N8042) was acquired from the Nottingham Arabidopsis Stock Center (*Browse, McCourt & Somerville, 1986*; *Scholl, May & Ware, 2000*).

Plant DNA was extracted by heating a small leaf piece (approximately $10 \text{ mm}^2$) in $750 \, \mu\text{L}$ extraction buffer (200 mM Tris–HCl pH 7.5, 250 mM NaCl, 25 mM EDTA, 0.5% SDS) for 5 min at 95 °C. The solution was left in room temp for a few hours before the DNA was precipitated through the addition of $750 \, \mu\text{l}$ 2-propanol. The supernatant was removed after centrifugation and the pellet washed once in 70% ethanol. Samples were dried in room temperature and the pellet containing DNA was reconstituted in $150 \, \mu\text{L} \, H_2O$.

PCRs was performed using a BioRad S1000 Thermal Cycler (Bio-Rad Laboratories, Inc., Hercules, CA, USA) with Titanium Taq polymerase (Clontech Laboratories, Inc., Mountain

**Table 1  List of primer sequences used for PCR.**

| Primer | Sequence |
| --- | --- |
| FAD7_FP1 | AAGACATAAGCGTGCGAACC |
| FAD7_FP2 | TGTTGCTAGTAGACCAACCCA |
| FAD7_FP3 | TACCTGCATCACCATGGTCA |
| FAD7_FP4 | AATCTCACATCACACCATCACT |
| FAD7_RP1 | TCAAAGCAGATTACACAGTTGCA |
| FAD7_RP2 | TTACCTTGCCACGGTACCAA |
| FAD7_RP3 | CTAACTCTCTGGTGGGTGACA |
| FAD7_RP4 | CGCACCTGGATCGAATCTCT |
| *pht4;1-3* _FP | CCACCTTTGGATCCTGCCTTTAT |
| *pht4;1-3* _RP | ATCAACAAACCACTGATTCAACTACACTT |
| CSHL_ DS5-2 | CCGTTTTGTATATCCCGTTTCCGT |

View, CA, USA) or AccuPrime *Pfx* DNA Polymerase (Invitrogen, Life Technologies, Carlsbad, CA, USA) according to manufacturers' instructions with primers listed in Table 1. PCR condition were as follows: 2 min initial denaturation at 95 °C, 30 s denaturation at 95 °C, 30 s annealing at 58 °C, 0.5–3 min extension at 68 °C (35 cycles), and final extension for 3–5 min. PCR products were analyzed by agarose gel (1–2% agarose; Seakem LE, Lonza, Switzerland) electrophoresis after staining with GelStar^TM (Lonza, Basel, Switzerland).

The transposon in *pht4;1-3* in crosses with *fad7-1* or L*er* were genotyped using gene specific primers pht4;1-3 _FP + pht4;1-3 _RP and transposon specific primers pht4;1-3 _FP+CSHL_ DS5-2 (Table 1).

### Extraction and quantification of lipids

The galactolipid species composition was analyzed as previously described (*Nilsson et al., 2014*) and total lipids fatty acid methyl esters was analyzed by GC-MS after direct transmethylation of leaf material. Leaf pieces were placed in boiling 2-propanol and supplemented with a known amount of di-nonadecanoyl phosphatidylcholine as internal standard. The samples were dried under a stream of nitrogen and transmethylated in 0.5 M sodium methoxide in dry methanol. The fatty acid methyl esters were extracted into heptane and analyzed by GC-MS as described (*Najm et al., 2017*).

## RESULTS AND DISCUSSION

Since PHT4;1 has been suggested to regulate phosphate transport inside the chloroplast (*Karlsson et al., 2015*), and since phosphate starvation is known to trigger exchange of phospholipids for galactolipids (*Andersson et al., 2003*), we profiled membrane lipids in the two *PHT4;1* mutant alleles *pht4;1-2* and *pht4;1-3*, and compared them to the parent wild type L*er* (Fig. 1). While there were no significant differences in the amount of the major membrane lipid classes, the *pht4;1-3* line displayed an altered species composition of several membrane lipids. This was clear for both major thylakoid lipid classes MGDG and DGDG (Fig. 1A), which demonstrated a substantial reduction in the highly unsaturated species 34:6 and 36:6. The *pht4;1-2* mutant line, on the other hand, did not differ from
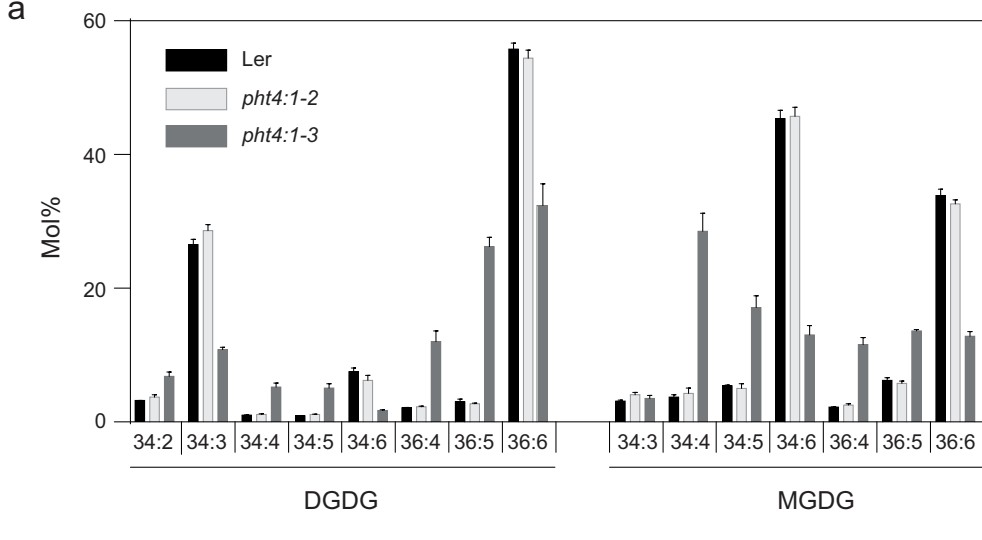

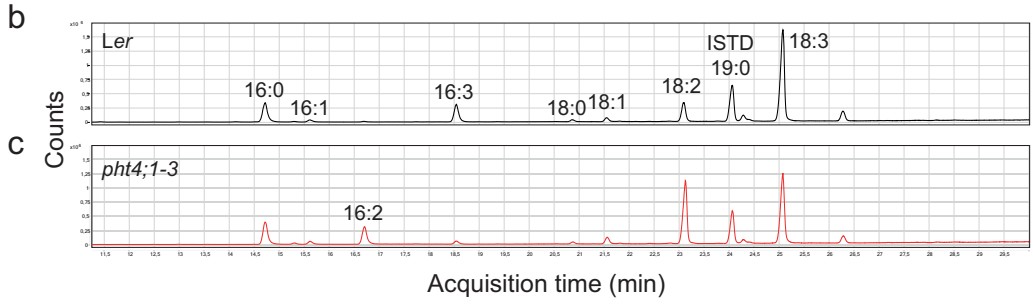

**Figure 1** **The *pht4;1-3* line has a deficiency in trienoic fatty acids.** Lipids were extracted from the indicated lines and subjected to LC-MS/MS and the species distribution of MGDG and DGDG is shown (A). Mean and standard deviation of four biological replicates are shown. Fatty acid methyl esters from wild type (L*er*) (B) and *pht4;1-3* (C) were analyzed by GC-MS and total ion chromatograms are shown.

wild type L*er* in its membrane lipid composition. Analysis of fatty acid methyl esters from a total lipid extract of the *pht4:1-3* mutant confirmed a decrease in linolenic- (18:3) and hexadecatrienoic (16:3) acids concomitant with an increase in the less saturated linoleic- (18:2) and hexadecadienoic (16:2) acids (Fig. 1C, Table 2). In particular, 16:3 was almost completely absent from the mutant line. This phenotype is very similar to that reported for mutants of the *FAD7* gene (*Browse, McCourt & Somerville, 1986*; *Iba et al., 1993*), leading us to suspect that a second site mutation affecting *FAD7* might be present in the *pht4;1-3* line.

To investigate this, we performed PCR on genomic DNA obtained from *pht4;1-3* and wild type L*er* using primers spanning different regions of the *FAD7* gene (Figs. 2A and 2B). A product containing the first exon and the 5′ UTR of *FAD7* was found to be absent in reactions from the mutant line (Fig. 2B). Further PCR with extended elongation time revealed an approximately 2.5 kbp insertion to be present in *pht4;1-3* in this region (Fig. 2C). Sequencing (sequence available in Data S1) of the insertion revealed it to contain an inactivated copy of the indole acetic acid hydrolase (IAAH) negative selection marker

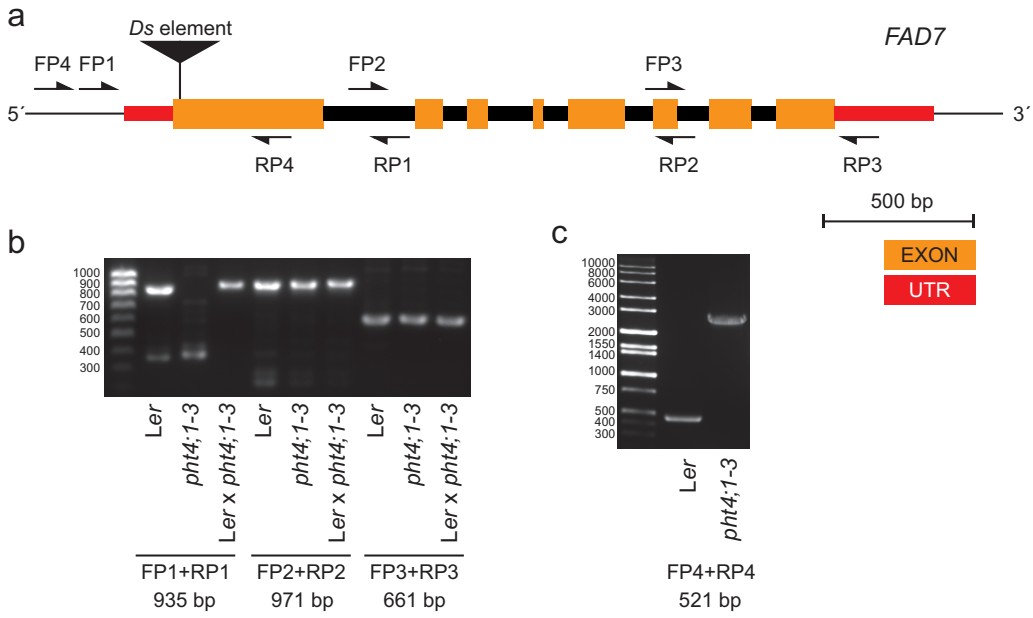

**Figure 2** The *pht4;1-3* line contains an insertion in the *FAD7* gene. (A) Gene model of *FAD7* and locations of primers used for PCR. (B) and (C) Agarose gel electrophoresis of PCR products from the indicated lines and primers. Expected fragment sizes are shown below lanes. An approximately 2.5 kbp insertion in *FAD7* in the *pht4;1-3* line is seen (C).

**Table 2** Fatty acid profiles of *pht4;1*-lines. Glycerolipid fatty acids were analyzed by GC-MS from leaf pieces obtained from the indicated lines. The following isomers are referred to for unsaturated fatty acids: 16:1, palmitoleic acid; 16:2, *all-cis* 9,12-hexadecadienoic acid; 16:3, *all-cis* 7,10,13-hexadecatrienoic acid; 18:1, oleic acid; 18:2, linoleic acid; 18:3, $\alpha$-linolenic acid. Mean and standard deviation are shown based on the total number of biological replicates as indicated in the table.

| Line | 16:0 | 16:1 | 16:2 | 16:3 | 18:0 | 18:1 | 18:2 | 18:3 | Replicates |
|------|------|------|------|------|------|------|------|------|------------|
| L*er* | 15 ± 0.9 | 1.5 ± 0.5 | 0.5 ± 0.1 | 12 ± 1.5 | 1.2 ± 0.4 | 3.1 ± 0.8 | 14 ± 2.0 | 57 ± 6.6 | 7 |
| *pht4;1-2* | 15 ± 0.2 | 2.0 ± 0.1 | 0.4 ± 0.1 | 14 ± 0.2 | 0.9 ± 0.1 | 2.2 ± 0.1 | 11 ± 1.1 | 55 ± 1.2 | 2 |
| *pht4;1-3* | 14 ± 0.9 | 1.5 ± 0.3 | 9.0 ± 1.8 | 2.0 ± 0.4 | 1.1 ± 0.3 | 4.3 ± 0.6 | 30 ± 1.6 | 38 ± 2.7 | 8 |
| *fad7-1* | 12 ± 0.2 | 1.4 ± 0.1 | 8.5 ± 0.1 | 1.7 ± 0.1 | 0.8 ± 0.1 | 4.1 ± 0.1 | 33 ± 0.2 | 39 ± 0.1 | 2 |
| *pht4;1-3, fad7-1*, F$_1$ | 12 ± 1.4 | 1.2 ± 0.1 | 8.4 ± 0.7 | 1.6 ± 0.1 | 1.2 ± 0.2 | 3.8 ± 0.3 | 32 ± 0.3 | 41 ± 2.2 | 2 |
| *pht4;1-3*, L*er*, F$_1$ | 16 ± 0.1 | 1.4 ± 0.1 | 1.2 ± 0.1 | 9.8 ± 0.1 | 1.6 ± 0.1 | 4.6 ± 0.4 | 17 ± 0.4 | 50 ± 0.9 | 2 |
| *pht4;1-3*, L*er*, F$_2$, *fad7-1* | 12 ± 0.3 | 1.5 ± 0.1 | 8.7 ± 0.1 | 2.0 ± 0.1 | 0.7 ± 0.1 | 5.1 ± 0.3 | 30 ± 0.7 | 39 ± 0.8 | 3 |
| *pht4;1-3*, L*er*, F$_2$, *FAD7* | 13 ± 0.2 | 1.6 ± 0.1 | 1.1 ± 0.8 | 10 ± 0.8 | 0.8 ± 0.1 | 3.2 ± 0.5 | 15 ± 2.1 | 56 ± 2.1 | 6 |

used to create the transposon line (*Sundaresan et al., 1995*). Further, the insertion was found to be located in the first exon of the *FAD7* gene (Fig. 2A, Data S1).

The *Ds* transposable element of the *pht4;1-3* starter line DsG6 has previously been mapped to the *FAD7* gene (*Parinov et al., 1999*). It thus appears that a part of the *Ds* element has remained in place in the *pht4;1-3* line due to the inability to use the negative selection based on IAAH. Rearrangements of the *Ds* element after transposase activation and subsequent inability to select against progeny carrying the starter line cassette have previously been reported (*Parinov et al., 1999*).

To confirm that that the altered lipid profile of *pht4;1-3* is indeed caused by the second insert in *FAD7*, a genetic test was conducted. The *pht4;1-3* line was crossed to *fad7-1* or wild type L*er* and fatty acid profiles of the resulting $F_1$ and parental lines were analyzed (Table 2). This clearly shows that the fatty acid desaturation defect in *pht4;1-3* is recessive and that the $F_1$ of *pht4;1-3* and *fad7-1* show an identical phenotype to their parents, supporting the notion of an inactivation of *FAD7* in the *pht4;1-3* line. Finally, *pht4;1-3* was backcrossed to L*er* and the resulting $F_2$ plants were genotyped using PCR primers for the insertion in *FAD7* and the previously described insertion in *pht4;1-3*. Selected $F_2$ plants homozygous for the two different insertions were tested for acyl lipid fatty acid composition. This fully supported the inferred linkage between loss of trienoic fatty acids and the *FAD7* insertion.

The *pht4;1-3* mutant line has, as far as we could find, only been used in two publications (Karlsson et al., 2015; Wang et al., 2011). Careful scrutiny of these reports shows that in fact all but one experiments are based on the use of at least one additional allele of *pht4;1*. The one experiment only including the *pht4;1-3* allele and wild type describes the spacing of thylakoids membranes in grana stacks (Karlsson et al., 2015). Since this is a phenotype that could at least partially be attributed to the physicochemical properties of the thylakoid lipids, it seems critical that this result is confirmed with a *bona fide pht4;1* mutant.

## CONCLUSIONS

To conclude, this study underlines the importance that every inferred phenotype-genotype relationship is confirmed by at least two independent genetic lines of evidence, i.e., through the use of several independent knock-out or silenced mutant lines, or by genetic complementation.

### Funding
This work was supported by the Carl Tryggers foundation. The funders had no role in study design, data collection and analysis, decision to publish, or preparation of the manuscript.

### Grant Disclosures
The following grant information was disclosed by the authors:
Carl Tryggers foundation.

### Competing Interests
The authors declare there are no competing interests.

### Author Contributions
- Anders K. Nilsson conceived and designed the experiments, performed the experiments, analyzed the data, wrote the paper, prepared figures and/or tables, reviewed drafts of the paper.
- Mats X. Andersson conceived and designed the experiments, performed the experiments, analyzed the data, contributed reagents/materials/analysis tools, wrote the paper, prepared figures and/or tables, reviewed drafts of the paper.

## Data Availability

The raw data has been provided as a Supplemental File.

## Supplemental Information

Supplemental information for this article can be found online at http://dx.doi.org/10.7717/peerj.4134#supplemental-information.

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
