# Peer review of "The pht4;1-3 mutant line contains a loss of function allele in the Fatty Acid Desaturase 7 gene caused by a remnant inactivated selection marker—a cautionary tale"

_PeerJ, doi:10.7717/peerj.4134_

## Round 0.1 · original submission · Major Revisions

There are concerns about the introduction of the manuscript and the details about the experiment itself that has to be cleared before acceptance. You will find details in the reviewers' reports here below.

Reviewer 1 ·

Basic reporting

The authors use of English is clear and professional, and there are some few minor recommended corrections that will further improve the clarity of the manuscript.

However, I think that the introduction is insufficient and more background needs to be provided to establish which is the gap in research that the authors aim to fill. The question being answered needs to be stated clearly in order to set up an appropriate context for the reader.

Experimental design

Please clearly state the question being addressed with your experiments.

Validity of the findings

No comment.

Additional comments

The authors address a problem often overlooked in reverse genetics. However, in my opinion the manuscript needs to state the question more specifically, as I found the introduction a little vague in that regard. Perhaps the authors would be better able to guide their readers by discussing the problem with establishing phenotypes of loss-of-function mutants, the need for researchers to ensure no other mutations in the background that might be causing them, and that having two or more alleles is preferable to define a
Aside from this, I recommend some minor corrections for clarity before resubmitting.
1. The major issue is that the introduction needs to be more specific about the aim of the study. One paragraph of introduction does not seem sufficient to lay out the background and the gap in research that this study means to fill.
2. Correct nomenclature of the genes and mutant alleles should be used consistently and clearly throughout the manuscript. For example, in line 36, PHT4 refers to the entire gene family (the correct gene name is PHT4;1). So it would be simpler to refer to “the transposon in pht4;1-3”
3. Line 34, change “were” to “was”.
4. Line 50, since the pht4;1-1 mutant allele was not mentioned elsewhere in the manuscript, I assume this is a typo and the authors meant to write “pht4;1-2 and pht4;1-3”. Please clarify.
5. Line 79, add a comma after fad7 and “was” after “conducted”.
6. Line 83, change to “F2 plants were”, instead of “F2 plants was”.

Reviewer 2 ·

Basic reporting

no comment

Experimental design

no comment

Validity of the findings

no comment

Additional comments

In this paper, the authors discribed a cautionary tale that the mutant line pht4;1-3 carried a remnant insertion in the FAD7 gene. And a series of biochemical and genetic evidences were supplied. The English laguage are professional and clear to understand. However, from this study, the authors concluded that every mutant should be confirmed by at least two independent lines. This is a well-known conclusion. Is it necessary to confirm again? Also, there are some major concerns as follows:
1. Introduction:
1) The introduction is too simple. Only PHT4;1 was discribed. More details about the phenotypes of pht4;1 mutants, FADs, as well as lipids and fatty acids should be introduced. Also, some mutagenesis methods should also be mentioned.

2. Materials and methods:
1) The mutant genotyping and the lipid analysis should be divided into two parts with two separate subtitles.
2) How did the authors get these mutant lines? From ABRC stock or given by the others? The stock numbers for these mutants?
3) How the Arabdopsis DNA was extracted? How did PCR and electrophoresis run?
4) Line 38: the tables should be listed in order. Table 1 is first, then folloed with Table 2, Table 3…
5) How many biological and technical repeats were carried? How many samples were included in each repeat?

3. Results and discussion:
1) Lines 50-51: the data about the lipid compositions are not shown?
2) Lines 53-55: in Fig 1, both DGDG and MGDG showed the quite different fatty acid compositions, why only MGDG was mentioned?
3) For all of the mutant lines mentioned in this paper, the authors should carry out the experiments such as Real-time PCR to detect the expression levels of PHT4 and FAD7.
4) The transposon was inserted into two separate positions (PHT4;1 and FAD7). This means more than one Ds copies in the mutant genome, maybe some other experiments such as southern blotting would be needed to detect the transposon insertion copies.

5. Tables and figures:
1) For the data in Fig1 and Table1, the significance analysis is required.
2) In Fig2, the insertion site in FAD7 should be marked in the gene model.

6. Supplemental data:
1) What did the capital letters and the lowercases mean in the FAD7 sequence? The blue letters are the coding sequence of FAD7 or not? If yes, the authors should indicate the triple coden, and the amino acid sequence should also be listed.

---

## Round 0.2 · accepted · Accept

There are a small number of very minor changes requested by Reviewer 1 which can be resolved while in Production.

Reviewer 1 ·

Basic reporting

No comment

Experimental design

No comment

Validity of the findings

No comment

Additional comments

The authors have satisfactorily expanded the introduction and added sufficient background as to aid in better understanding of the reasoning behind the experiments. I recommend only some minor changes to the manuscript for clarity:



1. Line 22: Reverse genetics
2. Line 52: phosphoglycerolipids
3. Line 77: change “were” to “was”
4. Line 100: “leading us to suspect” (delete “e” and “d”)
5. Line 105: revealed
6. Line 107: same as #5